# Co-Design, Delivery, and Evaluation of Wellbeing Initiatives for NHS Staff: The HOW (Healthier Outcomes at Work) NHS Project

**DOI:** 10.3390/ijerph19084646

**Published:** 2022-04-12

**Authors:** Jermaine M. Ravalier

**Affiliations:** Department of Psychology, School of Sciences, Bath Spa University, Bath BA2 9BN, UK; j.ravalier@bathspa.ac.uk

**Keywords:** wellbeing, working conditions, healthcare, intervention, app, stress, mental health, engagement

## Abstract

Stress and mental health are leading causes of sickness absence in the UK, responsible for over 50% of sickness absence across the country. Healthcare sector workers play a vital role in the life of everyone across the country but have among the highest levels of sickness absence of any sector. The aim of this project, therefore, was to work with UK healthcare workers to co-develop, implement, and evaluate a series of mental health and wellbeing interventions delivered via a smartphone app and associated toolkit. A participatory action research methodology, consisting of individual interviews, focus group discussions, and oversight by an expert action learning group, was used to develop primary (i.e., those associated with the workplace) and secondary (improving individual resilience and coping) stress management interventions. Pre-post-intervention analysis demonstrated improvements in work engagement and working conditions, although significant improvements were only found in mean scoring on demands, control, managerial support, and peer support working condition measures. The project therefore demonstrates that co-produced initiatives which focus on improving either the organisation or resilience of the workforce may be useful in supporting employee health and wellbeing. Future studies should build upon these findings through a full RCT to determine utility of the interventions.

## 1. Introduction

Stress in the workplace should be a key consideration for all employers in the United Kingdom (UK) because chronic workplace stress has been demonstrated to have potentially negative influences on employee physical and psychological health and wellbeing. It is therefore a key consideration for employers and employees alike. For example, the Interheart studies [1] demonstrated that chronic workplace stress has the potential to influence the development of cardiovascular disease to the same extent as already known risk factors, such as having high blood pressure and smoking. Similarly, Chandola, Brunner, and Marmot [2] found that chronic stress is related to the development of metabolic syndrome, which is a risk factor for the development of significant health complaints such as Type 2 diabetes. Chronic workplace stress has been associated with the development of psychological disorders such as depression and anxiety [3]. Interventions developed to support employee psychological health and wellbeing are therefore important, but often lacking.

### 1.1. Stress and Sickness Absence

Chronic stress and poor employee wellbeing not only affect the individual, but they also have knock on effects on their employing organization. In the UK, stress, depression, and anxiety are the leading causes of long-term sickness absence (that which lasts four weeks or more) and the second only to colds and flu for short-term absences [4]. They are therefore responsible for close to 18 million working days lost in the country, accounting for over half of all work-related ill-health cases [5]. This project therefore seeks to co-develop, disseminate, and evaluate a series of stress and wellbeing interventions for healthcare workers in the UK.

This paper will work within the framework of the job demands-resources (JDR) model of workplace stress to co-design, disseminate, and evaluate of series of mental health and wellbeing interventions for healthcare workers in the UK. There have been a number of workplace stress and wellbeing theories proposed [6]. The JDR suggests that the conditions under which employees work can be classified as demands, which add to the experience of stress, and resources, which buffer against these demand experiences. Demands add to the physical or psychological load and may include examples such as qualitative and quantitative workload. Resources, on the other hand, detract against the impact of these demands, and may include support (e.g., organisational or peer) and developmental opportunities [7]. Should demands outweigh available resources, then individual and organisational outcomes for sickness absence [8], job dissatisfaction, and burnout [9] may occur.

### 1.2. Working Conditions and Wellbeing

In a bid to support organisations in dealing with stress in the workplace, in 2004 the UK HSE released a set of management standards. These management standards are a set of working conditions, or psychosocial hazards, which, if left in a chronically poor state, can lead to adverse individual and organisational outcomes (e.g., sickness absence [10]). A review of the literature by the team who developed the management standards [10] identified seven distinct hazards (demands, control, managerial support, peer support, relationships, role, and change), and an associated survey tool (the Management Standards Indicator Tool; MSIT) was also developed to measure these. Poor levels of these working conditions have been associated with increased presenteeism [11], intentions to leave, and dissatisfaction [12] in several public and private sector organisations and sectors [13]. The management standards approach was initially based around the job demands–control–support (JDCS) model of stress, which suggests that it is the interaction of three components of the working environment (high demands, low control, and poor peer support), known as the iso-strain hypothesis, which leads to stress and related outcomes in the workplace [14].

Research is continually demonstrating the impact of poor working conditions on healthcare workers, their employing organisations, and ultimately on the care they provide to patients and service users. In the UK, health and social care workers have among the highest levels of stress and mental health sickness absence of any sector [5]. Indeed, these high levels of sickness absence are associated with high levels of qualitative [15] and quantitative workload [16], peer support, managerial support, and supervision [17]. among others [18]. Poor working conditions have also been shown to be related to poorer organisational outcomes such as job satisfaction, sickness presenteeism, and intentions to leave the role [12].

West and Dawson [19] demonstrated that clinical healthcare trusts in the UK which have positive levels of employee engagement had better patient morbidity and mortality outcomes, and better absenteeism and turnover rates among employees. Work engagement is defined as a positive work-related state which is characterised by vigour, dedication, and absorption [20], with engaged employees performing over an extended period. Vigour relates to high energy levels while working. Dedication describes a sense of being strongly involved in work and a sense of enthusiasm and challenge. Finally, absorption is characterised by being fully concentrated and engrossed in work [20]. Within the healthcare profession, employee engagement can be affected by organisational factors such as advocacy, patient bed availability [21], autonomy, employee wellbeing [22], and leadership [23]. In turn, better levels of employee engagement have been shown to positively impact patient safety [24], financial, and care outcomes [23], amongst others.

### 1.3. Psychological Health and Wellbeing Interventions

Interventions designed to support and improve the stress and mental health of employees at work are often characterised as primary, secondary, and/or tertiary approaches [25]. Primary approaches include interventions, such as job redesign and changes to work practices, and seek to influence the experience of stress by improving upon working conditions to reduce the source of stressors. However, these interventions can be both costly and time-consuming to implement and are therefore often overlooked. Secondary approaches aim to support employees to better cope with the stressors presented at work through greater coping and resilience. Examples may include cognitive behavioural therapy, mindfulness, and psychoeducation. These approaches are often utilised because they are quick to integrate and can be more cost-effective than primary approaches, but it is argued that they do not address the root cause of the issue. Tertiary approaches aim to support those who have already taken sickness absence to successfully return to work through approaches such as counselling and vocational rehabilitation. However, they do not provide support to employees who are at risk of sickness absence, nor address the root cause of this sickness [26]. A systematic review by Montano, Hoven, and Siegrist [27] found that organisational-level interventions can be supportive of both individual wellbeing and organisational-level outcomes, and a meta-analysis by Carolan, Harris, and Cavanagh [28] demonstrated that digital interventions can improve psychological wellbeing and organisational-level outcomes. However, both suggest that the quality of study published needs to be improved due to methodological weaknesses. This project aims to do just this: by working with healthcare workers, we will co-design, implement, and robustly evaluate a series of multi-level (i.e., those which aim to focus on more than one “type” of intervention, such as primary and secondary) digital psychological health and wellbeing interventions.

### 1.4. Project Aims

To sum, therefore, those working in UK healthcare have high levels of stress and mental health-related sickness absence, and the roles are replete with poor working conditions, leading to poorer psychological health and wellbeing and burnout, among other outcomes. The overall aim of this project was to work with healthcare staff in order to co-develop, implement, and evaluate a series of interventions to support and improve the working conditions, and psychological health and wellbeing, of workers. Secondarily, the aim is to investigate the influence of working conditions on psychological health and wellbeing of healthcare employees.

## 2. Methods

### 2.1. Design

This study undertook a pre-post methodology for the evaluation of a series of co-designed wellbeing interventions for NHS staff in two southwestern England employing trusts which employ approximately 6000 individuals. There are 217 NHS trusts in England and Wales, and they provide health and social care services to the general population. The individuals responsible for staff development and wellbeing in a number of southwest England NHS trusts were contacted by email by Ravalier to gauge interest, with four responding and forming the organisations participating within this project. During the co-design phases, though, two each of the trusts merged into two larger trusts. Ethical approval was gained from the Bath Spa University research ethics board; clearance code: JR181217.

Figure 1 below outlines the stages of data collection and intervention rollout. Interventions were designed using a participatory action research (PAR) methodology. PAR methodologies ensure that beneficiaries of a series of interventions play at least some part in the development, dissemination, and evaluation of those interventions [29], thus ensuring that mechanisms developed to support psychological health and wellbeing within this project were developed by NHS staff within participating organisations, and for themselves and their colleagues. Participants’ own expertise within their working situation is therefore used to develop a series of psychological health and wellbeing interventions, focusing on primary organisational interventions and secondary psycho-educational sources of support. As such, the researcher did not determine the interventions to be developed—they were developed through working with employees—and thus they emerged through the PAR process (i.e., interviews and focus groups, and overseen by an action learning group (ALG).

There were five separate but related sets of data collection, with an ALG meeting approximately every 3 months, and at the start of each phase of research, to provide continuing input into the project. The PAR process began with a series of developmental semi-structured interviews, with the findings of these informing several focus group discussions (see below). Following this, developed interventions delivered via a smartphone app and wellbeing toolkit were created, prior to being disseminated to all participating organisations for eight months (per NICE [30], guidelines). Evaluation consisted of pre- and post-intervention surveys. For each stage of data collection (pre-post-surveys, interviews, and focus groups), respondents were invited to take part in the study by all-staff emails sent on behalf of the research team by senior management within their organisation. Any and all responses, questions, and queries were directed to the research team, and not senior management, in order to maintain anonymity and confidentiality.

### 2.2. Materials

#### 2.2.1. Pre-Post-Intervention Surveys

All employees across the participating trusts were invited to complete the surveys via an all-staff email. The only inclusion criteria, therefore, were that they were employed within one of the trusts. As above, a pre-post survey of working conditions, perceived stress, and psychological wellbeing was undertaken as part of the evaluation strategy. Working conditions were measured using the 25-item version of the Management Standards Indicator Tool (MSIT [23]). The tool has been demonstrated to be sensitive for use in a number of frontline public sector roles, including the health service [31], police [11], social work [12], and teachers [32]. The MSIT measures seven areas of the workplace (demands, control, managerial support, peer support, relationships, role, and change communication) which, if left in a chronically poor state over an extended period of time, may have negative impacts on psychological and/or physiological health of employees. Respondents answer on a 1–5 Likert scale, and psychometric analysis demonstrated validity, reliability, and fit of model as being as acceptable as the originally developed 35-item version, with the added advantage of being quicker to complete [13].

Psychological health was measured firstly using the 14-item Warwick–Edinburgh Mental Wellbeing Scale (WEMWBS [33]) which uses 14 positively-phrased questions, answered on a five-point Likert scale, and asks about experiences over the past two weeks. The measure has been shown as preferential for use with health service personnel populations [34], and has been shown to have strong reliability and validity in a range of populations [35]. Lastly, the WEMWBS is sensitive to change at both group and individual levels [36].

Secondly, the four-item Perceived Stress Scale (PSS-4) was used to measure levels of perceived stress within the population. The PSS-4 assesses the frequency of stressful situations over the past month and is assessed via a four-item Likert scale. Again, psychometric properties have proven to be sound [37]. The MSIT, WEMWBS, and PSS-4 have each been shown to have utility when used together within one survey [12].

Three further measures of organisationally focused outcomes were also utilised in the project. Work engagement was measured using the nine-item version of the Utrecht Work Engagement Scale (UWES [9]). The UWES measures three components of work engagement: vigour, dedication, and absorption, with the nine-item version of the measure inherently valid and reliable, with good longitudinal reliability [38]. Responses are given on a seven-point Likert scale from 0 (never) to 6 (always), determining the frequency of work-based feelings. Job satisfaction and presenteeism were measured using two single-item global tools, which are argued to be as reliable as multi-item measures with the added advantage of being quick to administer and complete [39]. The job satisfaction measure asked, “Taking everything into consideration, how do you feel about your job as a whole?”, with responses on a five-point Likert scale [39]. Presenteeism was measured on a four-point Likert scale via the question “As far as you can recall, has it happened over the previous 12 months that you have gone to work despite feeling that you should really have taken sick leave due to your state of health?” [40]. Finally, demographic questions asked were age, gender, whether the individual identified as having a disability, ethnicity, and hour disparity. Hour disparity demonstrates a difference between the number of hours contracted to and the number of hours typically worked in a given week. In order to ensure anonymity and GDPR compliance, no data were collected through or about use of the app.

#### 2.2.2. Participatory Action Research (PAR) Methodology

In order to develop the interventions associated with this project, the PAR process consisted of a series of semi-structured interviews, followed by a number of focus group discussions, and the regular input of an ALG approximately every three months across the project, as well as ad hoc meetings at the end of each phase of research. Twenty individual semi-structured interviews were therefore undertaken with healthcare workers employed within the participating trusts. The interview schedule sought to determine the sources of organisational stressor associated with their work, role, and employing organisation, resources available to cope with these stressors, and interventions/changes which needed to be made in order to overcome these stressors by reflecting upon their ideal support, and support that they had seen, experienced, or heard about in other organisations (see [18]). Interviewees were recruited through an all-staff email sent by senior management in each trust and conducted on a sequential basis. Iterative rounds of data collection and analysis continued through to saturation point across all participating organisations [41]. No job roles were excluded from interviews, which lasted on average 45 min each and occurred between February and May 2018, and all were undertaken by Ravalier over the telephone and digitally recorded. Written consent was taken at least 48 h prior to interviews, as well as verbal consent at the beginning of interviews, and participants were debriefed at the end.

Following completion of interviews, four mixed focus groups of employees from various job roles and levels of seniority were undertaken. Lasting approximately 90 min each, participants were again recruited via all-staff emails designed by the project team and sent by senior management. Similar to Ravalier et al. [31], the aim of these focus groups was to elaborate upon, further develop, and ensure the feasibility of the suggested interventions from the individual interviews. Due to the potential geographic distances and differing shift patterns between participants, focus groups were run virtually using the “Go to Meeting” platform. Focus groups began with the researcher setting the ethical ground rules for the interview, and each meaningful intervention from individual interviews were discussed. Emergent points were listed on the virtual whiteboard and revisited at the end of the focus group to ensure member-checking of suggested interventions and approaches. Consent was gained in the same method as the individual interviews.

ALGs were held approximately every three months, and at the end of each project stage. This group consisted of 13 key stakeholders taken from across each organisation, with members including senior management, staff, and union representatives. The three-monthly meetings were undertaken to allow for the group to provide feedback, shape and guide the progress of the project. Meetings held at the end of each phase of research subsequently allowed input into the findings of the previous stage as well as the aims/approaches taken in the following stage. No participants were paid for their contribution to the ALG, but senior management in each trust agreed that they would be able to be run during working hours with no expectation of having to work extra hours to make up for work missed. ALGs were undertaken in person, but at different locations in order to ensure that as many people as possible could make the physical meetings. ALGs also helped to design the physical “look and feel” of interventions, and intervention delivery method.

Interventions were disseminated through a series of approaches. Firstly, all-staff emails were developed by the research team and ALG groups. Furthermore, a set of three events lasting four hours each (broken into shorter 30 min sections) provided information to potential users on how to download and use the app/toolkit and interventions, with food and drinks provided. Invitations (and information about the project) were shared via all-staff emails, specifically developed newsletters, meetings with senior and middle management (who then shared information), and attendance/presentations at staff meetings.

### 2.3. Analytical Method

#### 2.3.1. Quantitative Analysis

Descriptive statistics (means, standard deviations, frequencies where appropriate) were first calculated for all measures (see Table 1). Subsequently, independent samples *t*-tests were undertaken to investigate differences in all measures pre- and post-intervention, and multiple linear regression analyses were conducted on post-intervention data to investigate the influence of working conditions, job satisfaction, and presenteeism on WEMEBS, PSS-4, and UWES outcome measures. All quantitative analyses were conducted using Jamovi (available at www.jamovi.org; accessed on 1 February 2022), with statistical significance set at <0.05. Evaluation occurred approximately 7 to 8 months after intervention roll out across all organisations. Evaluation consisted of pre- and post-evaluation all-staff surveys at a group (mean) level.

#### 2.3.2. Qualitative Analysis

Both sets of semi-structured interviews were analysed using a data-driven thematic analysis approach [42] which seeks to draw general themes from data such as semi-structured interviews, and thus allowing the identification, analysis, description, and reporting of these themes. We undertook the six-step approach by Braun and Clarke [42], following transcription by a GDPR-approved external organisation and anonymisation by the lead researcher. Focus groups were self-analysed and member-checked per the above. 

## 3. Results

### 3.1. PAR Intervention Development: Semi-Structured Interviews

The aim of these semi-structured interviews was to determine what intervention strategies colleagues from within the participating organisations feel would support their wellbeing, either by changing the work environment or providing greater support. The interview schedule therefore opened with questions about respondents’ job and organisation (as a form of rapport builder), about the current stressors, wellbeing initiatives offered within the organisation and how these can be improved, whether they had experienced or heard about initiatives in other organisations that would be supportive in their current role/organisation, and what other initiatives and interventions they would like to see within their organisation to support staff wellbeing. These twenty interviews were analysed thematically, focusing on suggested interventions, with interviewing continuing through to saturation.

#### 3.1.1. Primary Interventions

##### Work Practices and Approaches (Peer Support)

The most frequently and passionately discussed approach to supporting and improving work practises focused upon formalising and diversifying support that individuals received from their colleagues. Every respondent suggested that they received excellent support from the peers around them, and that this support could be broadened to more clearly support their psychological wellbeing at work. This includes peer support programmes which meant that individuals received support from those within their department, as well as formalising support mechanisms between departments to support the learning of individual employees and the department.
*“When there are intrinsic links between teams, maybe there could be an individual within each team who almost takes a little bit of responsibility for spreading some of those messages.”*.(Interviewee 7, non-clinical)
*“I think it (formalised peer support) would help because it would give someone the opportunity to talk to someone who’s not in the situation who can give a balanced and outside view.”*.(Interviewee 10, clinical)

##### Bottom-Up Communication

Communication, both bottom up and top down, was often discussed as a source of difficulty by respondents. Bottom-up communication was discussed as a lack of opportunity to inform organisational, departmental, and even individual roles. In particular, respondents suggested that they had previously attempted to provide organisational or departmental methods for improvement to senior management, but these were often overlooked and ignored. Alternatively, where suggestions for change could be seen as being critical of the organisation, participants suggested that they had no confidential method of feeding constructive feedback to senior management. Therefore, interviewees described wanting a communication methodology that was anonymous (if necessary) while also ensuring that at least some communication was at least acknowledged.
*“And if you try to pass anything back up the chain, it disappears into the ether then when it gets beyond this wall.”*.(Interviewee 1, non-clinical)
*“Well, if you emailed, you do email with your own personal mail so that wouldn’t be confidential.”*.(Interviewee 3, clinical)
*“Going through and sending forward best practice ideas up through team managers, up into the clinical governance meetings. Even sending things into the newsletter about feeding up all that information and having at least some of that, erm, that responded to would be helpful.”*.(Interviewee 8, clinical)

##### Top-Down Communication

Top-down communication (i.e., that from management to staff), particularly with respect to employee wellbeing, was similarly discussed as an issue. Across the different trusts, employees worked out of a spread of geographical locations (i.e., while some would be based in a hospital within the county, others would be based within an administrative building which was some distance away). This meant that respondents suggested they did not know whether they were allowed to attend activities that were happening at different locations, unsure as to who was running events and how to contact them, unsure where to look for information, and many described the invitations to events as being quite last-minute. Furthermore, others wanted more specific methods of communication to different directorates and departments, rather than organisation-wide news and briefs.
*“Things that come from about our level, it’s like there’s a big wall goes up, and the little bits that drip down to us.”*.(Interviewee 1, non-clinical)
*“We used to get some more detailed information through the trust newsletter, but we found that there’s too much hard work waiting for them now because it’s all focused on [name of service].”*.(Interviewee 5, non-clinical)
*“I think our weekly trust bulletin is one size fits all, which clearly doesn’t really work. Maybe we can get more specific directorate bulletins and target it at specific levels, I think.”*.(Interviewee 10, clinical)

#### 3.1.2. Secondary Interventions

##### Psychoeducation

Psychoeducation has been demonstrated to be supportive of wellbeing at work, in particular as a method of early intervention. Respondents suggested this could be an important addition to both the app and toolkit because it would allow themselves and peers to refer to appropriate support before Individuals became too ill.
*“I think in a first-level, I know that I’d go quiet if I’m worried about something. But I, erm I’m not great at introspection–looking inwards–often until it’s too late.”*.(Interviewee 19, clinical)
*“It’s hard isn’t it? Like, I think I–I can tell when my close friends and colleagues are struggling I think, hope. But it’s harder seeing in myself.”*.(Interviewee 17, non-clinical)

##### Wellbeing-Related Activities

All interview participants appreciated that their employing trust made available a series of wellbeing-related activities. These included group walks and other group physical activities, an employee assistance programme, counselling and support, and other related activities. However, due to the geographically diverse nature of the organisations, and shift work often associated with both clinical and non-clinical work, many argued that it was very difficult to either access or be part of some of these activities.
*“The problem is if those things do exist in parts of the county, it does leave the feeling- an area of neglect. The area is still—I think [removed for anonymity], we do feel a little bit out on a limb because things don’t really seem to get out our way.”*.(Interviewee 8, clinical)

#### 3.1.3. Other Interventions

In addition to support mechanisms which support both primary and secondary level interventions, a number of suggestions were made which may have been supportive of colleague mental health and wellbeing but were likely unfeasible or unpractical for financial reasons. These include development of cycle paths and purchase of organizationally-held bicycles for use during breaks, poor physical working conditions, such as cramped workspaces for both clinical and non-clinical staff, and the provision to spend a day or two per year in order to work within another organisation, such as a charity.
*“I would love the cycle paths that I’ve seen talked about over at [removed for anonymity].”*
*“You can go and sit on a bike and bike for 15 min.”*.(Interviewee 8, clinical)
*“You literally haven’t got the space. It’s a really cramped working space, there could be 15 people in the room so it’s quite noisy, it’s a really difficult working space. We need more space.”*.(Interviewee 9, clinical)
*“In the private sector, they offer you one day per year that you can go and visit a charity and work, be paid for the day working for that charity. That’s quite a good idea and it gets you out and about in understanding things.”*.(Interviewee 10, clinical)

### 3.2. PAR Intervention Development: Focus Groups and Intervention Contents

The aim of the focus groups was to turn ideas and suggestions from interviews into feasible and pragmatic interventions which could be disseminated across organisations via an app and toolkit. Therefore, the discussion schedule following interviews was focused upon peer support, communication, psychoeducation, and wellbeing-related activities, and how these could be presented. These suggestions, once input into a prototype app and toolkit, were then brought to the ALG groups for feasibility assessment. App look, feel, and content could be easily updated by the research team using a dedicated online content management system website.

Peer support, which was both formalised and increased in depth, was agreed by focus group participants to be a key method of supporting wellbeing. Through the app, users could request a peer support buddy. This buddy likely did not work within the same department, nor in the same role, but the pairs acted as peer support for each other. Furthermore, it was widely understood that different people, even within the same department, had different interests and expertise. Focus group respondents wanted the opportunity to not only allow colleagues the opportunity to demonstrate their knowledge and expertise, but also provide support in these expert areas to colleagues. Therefore, through the toolkit, a blank and example “support tree” was presented. In this tree, line managers could present colleagues’ interests and expertise as branches. For example, in a clinical setting, some colleagues may have had expertise in equality, diversity, and inclusivity (EDI). This would therefore be a named person, in a poster available both electronically and physically, that colleagues could go to should they have EDI issues or queries.

In order to improve upon bottom-up and top-down organisational communication, focus group respondents wanted a reporting mechanism, available through the app. Users could propose constructive and positive organisation changes, either anonymously or otherwise, which would be emailed directly to a prearranged senior manager’s email address within each organisation. The senior management team in each organisation would then respond to at least three of these suggested changes in all-staff newsletters. Focus group participants suggested that this was one way of having colleagues’ voices heard and ensuring that management either act on suggestions or explain why they cannot act on these suggestions.

Early identification of wellbeing-related strain in themselves and others was mentioned as an important, but difficult, area to support. As such, while some seemed to know triggers and signs of deteriorating wellbeing in themselves, they did not necessarily know what to look for in others, and vice versa. The psychoeducational component of the app and toolkit both therefore outlined signs and symptoms of strained mental and psychological health in individuals, colleagues, and, more widely, teams, based initially upon the UK Health and Safety Executive advice (HSE, n.d.). Furthermore, advice on healthy work and working practices were presented.

Finally, it was appreciated that there were wellbeing-related activities and support mechanisms happening and available across all participating organisations. However, it was unclear where all of the information relating to wellbeing in the different organisations were stored. Respondents therefore wanted the app to hold all information on mental health and wellbeing offerings available, whether that be events, organisational support such as employee assistance programmes, or something else. For events that happened across the different organisations, both the research team and wellbeing leads within the organisations could update the app and send push notifications whenever there was related activity or changes to promote. Similarly, the toolkit, which was designed to be bespoke for each participating organisation, also held details of all wellbeing-related organisational support available.

### 3.3. Quantitative Analysis

#### 3.3.1. Descriptive Statistics

Table 1 outlines demographic characteristics for survey participants at pre- and post-intervention (Time 1 and Time 2, respectively). The mean age of respondents was similar across the two time points, with small differences across gender, whether individuals identified as having a disability, and ethnicity. Hour disparity demonstrated an average of 3.49 h worked per week more than contracted in Time 1, and 1.89 in Time 2.

Table 2 demonstrates mean and standard deviation scoring for the WEMWBS, UWES (and constituent factors), and MSIT (and its seven factors). Other than for the absorption measure within the UWES, all mean and percentage scores were increased at Time 2 when compared to Time 1. A series of independent samples *t*-tests was undertaken to determine whether differences between the two time points were significant. Significant differences were found between the two times in demands (t = −5.00, *p* < 0.001), control (t = −2.66, *p* < 0.005), managerial support (t = −3.08, *p* < 0.005), and peer support (t = −2.28, *p* < 0.05), with change near significant (t = 1.90, *p* = 0.06). All differences represented positive changes in working conditions. No significant differences were found across time between “relationships” and “role” of the MSIT, mean score of the PSS, the UWES factors (mean score as well as vigour, dedication, and absorption), and the mean WEMWBS. Percentage frequency scores for presenteeism and percentage dissatisfied were similar between the two time points.

#### 3.3.2. Regression Analyses

Table 3 demonstrates the findings from a series of regression analyses looking at the impact of factors relating to working conditions (MSIT), presenteeism, and job satisfaction, on psychological wellbeing (WEMWBS), perceived stress (PSS-4), and work engagement (UWES) at Time 2. All three models were significant (*p* < 0.001), and for each variable in each model VIF is above 0.2, and tolerance less than 10, indicating no collinearity (Field, 2013). The first model (WEMWBS) accounted for 33% of the model variance and each of control, peer support, role, job satisfaction, and presenteeism were significantly associated. Coefficient estimates suggest that peer support was the greatest influence on WEMWBS scoring, with greater reported peer support meaning improved psychological wellbeing. For model two (PSS-4), the four significantly related variables (demands, peer support, job satisfaction, and presenteeism) accounted for 29% of the variance, with the experience of presenteeism the greatest influence on perceived stress. A total of 45% of the variance was explained in model 3 (UWES) with significant associations, with each of demands, control, relationships, role, change, job satisfaction, and presenteeism, and job satisfaction being most impactful on engagement.

## 4. Discussion

### 4.1. Findings

The aim of this project was to work with healthcare workers, employed in UK LAs, to co-develop, disseminate, and subsequently evaluate the impact of a number of wellbeing interventions, disseminated through a smartphone app and associated toolkit. Through a PAR process of semi-structured interviews and focus groups, as well as reflective and confirmatory ALG groups, a series of interventions were co-developed. The PAR process also developed a dissemination strategy in an attempt to ensure maximum take-up and engagement with the work.

Interventions were developed through the PAR process and, in particular, the outcomes of the interviews and focus groups. The ultimate aim of the interviews was to develop an understanding of the types of intervention which could be developed and utilised within the project, with the focus groups then developing pragmatic and workable methods of implementing these. Interventions were primary (i.e., changing organisational or working practices) or secondary (i.e., supportive the psychological wellbeing of employees). Primary interventions included changes to work practices by providing greater peer support. Secondly, top-down and bottom-up communication strategies were developed, which included communication around wellbeing events and strategies, as well as organisational development feedback mechanisms. Secondary interventions were psychoeducational in nature, thus aiming to support understanding of wellbeing at work. This included information on the signs and symptoms of poor mental health, organisational sources of support, and healthy living/working advice.

Evaluation of the work was undertaken via pre- and post-intervention surveys. Significant differences between pre- and post-intervention surveys were demonstrated in the demands, control, managerial support, and peer support working conditions. Considering the primary and secondary nature of the interventions developed, these significant improvements may not be surprising. The primary interventions developed sought to make changes to organisational or work practices, and thus likely led to improvements in workload. For example, by developing and sharing new ways of working across teams, this likely reduced the amount of administration required of social work teams. Indeed, peer support was a key component of primary interventions. By working with colleagues to develop new ways of working, this will have led to more control over the way in which work was conducted. Similarly, improvements in managerial support may have emerged due to greater bottom-up and top-down communication.

Despite this, while most other variables measured showed differences between pre- and post-intervention, none of these were significant. One potential reason for this is the COVID-19 pandemic. Post-intervention surveys were undertaken during the pandemic, which affected the approach to work of many healthcare workers [43]. Respondents already scored within the average range on work engagement too, and thus further improvements in engagement may be difficult to ascertain. Regression findings, which aimed to demonstrate impacts of working conditions on psychological wellbeing, stress, and engagement, support those of the pre-post-intervention analyses. No single management standard-assessed condition was significantly impactful on all three outcome measures. However, control, peer support, and role understanding each impacted two of the three outcomes. In addition, both job satisfaction and presenteeism significantly influenced each of wellbeing, stress, and engagement.

The iso-strain interaction of the JDCS suggests that chronic exposure to the combination of high demands, low control, and poor peer support leads to negative stress and related outcomes at work [14]. Findings from the pre-post-intervention demonstrated that two types of support (managerial and peer), control, and demands were improved, as were measures of wellbeing (although not significantly). At post-intervention, it was also demonstrated that control and peer support significantly impacted outcomes. These findings mirror those of researchers such as Van Der Doef and Maes [44], and Hassuer et al. [45], who reviewed literature from 1979 through to 2017 and demonstrated evidence for the iso-strain hypothesis. Similarly, Wilberforce et al. [46] demonstrated that the interaction between high workload and poor autonomy was related to greater dissatisfaction.

Various projects have sought to use PAR methodologies for the development of work stress and wellbeing interventions in healthcare populations. For example, McVicar and colleagues [47] demonstrated it as a useful methodology of interventional design, and Dollard et al. [48] discuss some methodological difficulties. However, few studies have evaluated the efficacy of such approaches on working conditions and psychological health. Despite this, Lenthall et al. [49] did utilise PAR to develop organisational interventions and did not show any improvements. Richardson and Rothstein’s [50] meta-analysis demonstrated that while secondary (and in particular relaxation) stress management interventions were the most frequently used, primary interventions were less so, although described as likely the most effective, with a mixture of primary and secondary approaches being preferable. This project therefore utilised a mix of primary and secondary approaches, and again saw improvements in some working conditions. Lastly, smartphone apps for use in the workplace as a method of disseminating interventions have shown promising results. For example, Weber et al. [51] demonstrated improvements in stress and wellbeing over a six-week period when compared to a non-user control group. However, unlike the current project, the project measured impact over a short period (6 weeks) and used secondary-only (psychoeducational) interventions.

Research across the pandemic has generally shown that the wellbeing of frontline workers has deteriorated as the pandemic began. For example, McFadden et al. [43] found that between May and November 2020, both work-related quality of life and wellbeing in health and social care workers worsened. Other studies have demonstrated that increased working hours during the pandemic were associated with poorer working conditions and psychological health [52], although having a positive attitude toward the excess stress experience during the pandemic is a positive coping strategy [53]. Working through the pandemic also had significant effects on the work–life balance of healthcare workers, with this subsequently impacting their wellbeing [54]. Despite these obvious effects of the pandemic on the wellbeing of these key frontline workers, our project found improvements (although not significant) in wellbeing when it would otherwise be expected for wellbeing to have become worsened.

### 4.2. Strengths and Limitations

As with any study, there are distinct strengths and limitations which need to be addressed. As noted above, this is one of the first longitudinal studies which has used a PAR methodology to co-design psychological wellbeing interventions, and subsequently evaluate with a pre-post-intervention survey. However, causation cannot be ascertained here. No control group was utilised, and the evaluative approach was not a randomised controlled trial. Furthermore, there are relatively small response rates at both pre-and post-intervention, and only approximately 10% of employees “used” or were aware of interventions. This must be improved upon in future studies, by maximising marketing of interventions within and across organisations, and divesting the range of marketing approaches used. Despite this, it is expected that there would be significant intervention “bleed” from interventions users to the rest of staff, in particular when using peer support and psychoeducation to support the health and wellbeing of others. Furthermore, the small response rate may be at least in part to the COVID-19 pandemic. A further limitation is that, because the app and toolkit could be personalised by organisation, managers, and individuals, and the app contained a number of different interventional approaches, there is no way to determine which intervention was most impactful over any other. However, as noted above, multi-component interventions are often most impactful, and therefore this approach is useful. Finally, there was significant study attrition from Time 1 to Time 2. As such, it is not possible to take into account individual-level effects which may have impacted the outcomes of the study. In particular, while interventions in the study were rolled out across different organisations, and within different departments in these organisations, the potential differences across these organisations and departments were not accounted for in the data collection and analysis. This therefore means the findings of the work should be taken with caution.

### 4.3. Implications and Future Research

Future studies should focus upon further robust evaluation in pilot or full-scale RCTs. While this project demonstrated potential utility of both the PAR approach and co-developed interventions, the pre-post methodology meant causation could not be ascertained. Randomising across a national sample would therefore support more robust evaluation of the project. Furthermore, we have shown increased utility of the JDCS model of stress, and this should again form part of the make-up of any future projects. The COVID-19 pandemic has led to a decrease in the working conditions and wellbeing of health and social care workers in the UK and beyond, and little support is available to protect these important workers from the impacts of the pandemic [43]. Future research should look to work with health and social care employees to tailor these interventions to support such workers post-pandemic. This may include approaches such as improved peer support and greater support for those working from home and more widely. However, this should be developed alongside health and social care workers through a further PAR methodology.

## Figures and Tables

**Figure 1 ijerph-19-04646-f001:**
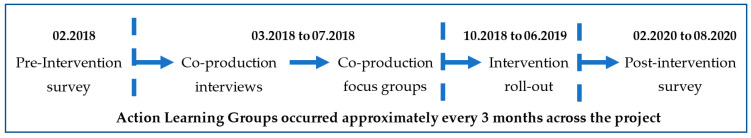
Project methods, timeline, and post-intervention survey.

**Table 1 ijerph-19-04646-t001:** Participant demographics pre- and post-intervention.

	Age (Mean, SD)	Gender (%)		Disability (%)		Ethnicity (%)	Hour Disparity (Mean, SD)
		Male	Female	Yes	No	White British	
Time 1 (*n* = 786)	45.4 (11.1)	14%	85%	6%	94%	91%	3.49 (5.24)
Time 2 (*n* = 129)	44.9 (12.9)	8%	92%	14%	86%	96%	1.89 (4.21)

**Table 2 ijerph-19-04646-t002:** Descriptive statistics for all measures used within the project pre- and post-intervention.

		Time 1	Time 2
Warwick–Edinburgh Mental Wellbeing Scale Mean (SD)		46.49 (9.88)	47.14 (11.19)
Perceived Stress Scale Mean (SD)		2.73 (0.83)	2.64 (0.86)
Utrecht Work Engagement Scale	UWES Mean (SD)	3.87 (1.07)	3.95 (1.05)
	Vigour Mean (SD)	3.36 (1.27)	3.57 (1.23)
	Dedication Mean (SD)	4.18 (1.17)	4.29 (1.17)
	Absorption Mean (SD)	4.09 (1.09)	4.02 (1.04)
Management Standards Indicator Tool	Demands Mean (SD)	3.37 (0.87)	3.77 (0.81) ***
	Control Mean (SD)	3.40 (0.82)	3.61 (0.81) **
	Managerial Support Mean (SD)	3.55 (0.87)	3.80 (0.84) **
	Peer Support Mean (SD)	3.88 (0.68)	4.02 (0.72) *
	Relationships Mean (SD)	4.32 (0.74)	4.38 (0.73)
	Role Mean (SD)	4.13 (0.69)	4.17 (0.68)
	Change Mean (SD)	3.06 (0.84)	3.20 (0.82)
Presenteeism (Frequency)		43.6%	40.7%
Job Satisfaction (Percentage Dissatisfied)		23.9%	23.0%

* Significant at <0.05, ** significant at <0.005, *** significant at <0.001.

**Table 3 ijerph-19-04646-t003:** Regression analyses of the impact of working conditions on psychological wellbeing, perceived stress, and work engagement.

	Significantly Related Factors	Coefficient Estimates	*t*	*p*	Tolerance	VIF	R^2^	Adjusted R^2^
Warwick–Edinburgh Mental Wellbeing Scale	Control	0.97	2.58	<0.01	0.78	1.28	0.34	0.33
	Peer Support	2.77	5.57	<0.001	0.64	1.56		
	Role	1.15	2.54	<0.05	0.76	1.31		
	Job Satisfaction	1.88	6.83	<0.001	0.57	1.74		
	Presenteeism	−2.27	−7.22	<0.001	0.85	1.17		
Perceived Stress Scale	Demands	−0.13	−4.19	<0.001	1.27	0.78	0.29	0.29
	Peer Support	−0.12	−2.93	<0.005	1.51	0.66		
	Job Satisfaction	−0.14	−6.38	<0.001	1.63	0.61		
	Presenteeism	0.24	8.78	<0.001	1.22	0.82		
Utrecht Work Engagement Scale	Demands	−0.13	−3.56	<0.001	0.74	1.35	0.45	0.45
	Control	0.15	3.91	<0.001	0.71	1.40		
	Relationships	−0.07	−1.77	<0.01	0.75	1.34		
	Role	0.13	2.89	<0.005	0.68	1.48		
	Change	0.18	4.22	<0.001	0.53	1.89		
	Job Satisfaction	0.41	15.43	<0.001	0.57	1.76		
	Presenteeism	−0.07	−2.22	<0.05	0.79	1.26		

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
