# Peer review of "Co-Design, Delivery, and Evaluation of Wellbeing Initiatives for NHS Staff: The HOW (Healthier Outcomes at Work) NHS Project"

_ijerph, 2022, doi:10.3390/ijerph19084646_

Round 1

Reviewer 1 Report

The paper debates a really relevant and delicate topic, which is really important to bring up to the research literature.

Below few advice I hope could be helpful to improve its quality

  • I would suggest to further develop section 1.1. Stress and Health with a wider explanation
  • I would suggest to further develop section 1.3. Psychological Health and Wellbeing Interventions, providing a synthesis of pros and cons of each type of interventions
  • I would suggest to add further details behind the process management and decisions (for examples, which kind of criteria/reasons were applied to define the type of intervention? were followed specific criteria to identify the people to engage?)
  • I would suggest to stronger highlight the connection and interpretation of qualitative and quantitative data - moreover, thinking about future possibile projects based on this paper, is it possible to hypothesize any kind of integration/modification of the intervention presented based on the pandemic and its effects?

Author Response

Thank you for your review and comments.

Please see attached your comments and my repsonse.

Author Response

Thank you for your review and comments.

Please see attached your comments and my response.

Reviewer 3 Report

Sorry for the delay in responding. I have been contemplating what to say.

Mostly, I think this is a paper on a topic of great importance using an approach that will  stimulate practitioners and researchers to think through hw they can do intervention research that positively impacts on health workers 

Before this could be considered for publication, the issue of referencing needs to be clarified. A number of times the reference is “(removed for anonymity).  Occasionally I can see the reason but not always. This is very odd and you have to decide if this is acceptable or not. It does mean the comments cannot be verified. If this is not acceptable there will need to be some rewriting.

The article needs to be proofread, as here as mistakes with the words chosen in some places.

Author Response

(The authors gave the same response as above.)
